

# Unravelling Landslide Failure Mechanisms with Seismic Signal Analysis for Enhanced Pre-Survey Understanding

Jui-Ming Chang[1,2], Che-Ming Yang[3]*, Wei-An Chao[1,2], Chin-Shang Ku[4], Ming-Wan Huang[3,5], Tung-Chou Hsieh[2], Chi-Yao Hung[6]

[1]Department of Civil Engineering, National Yang Ming Chiao Tung University, Hsinchu 30010, Taiwan

[2]Disaster Prevention and Water Environment Research Center, National Yang Ming Chiao Tung University, Hsinchu 30010, Taiwan

[3]Department of Civil and Disaster Prevention Engineering, National United University, Miaoli 36063, Taiwan

[4]Institute of Earth Sciences, Academia Sinica, Taipei 11529, Taiwan

[5]He Yu Engineering Consultants Co. Ltd., Taichung 40642, Taiwan

[6]Department of Soil and Water Conservation, National Chung Hsing University, Taichung 40227, Taiwan

*Correspondence to*: Che-Ming Yang (stanleyyangcm@nuu.edu.tw)





## Abstract

Seismic signals, with their remote and continuous monitoring advantages, have been instrumental in unveiling various landslide characteristics and have been widely applied in the past decades. However, a few studies have extended these results to provide geologists with pre-survey information, thus enhancing the understanding of the landslide process. In this research, we utilize the deep-seated Cilan Landslide (CL) as a case study and employ a series of seismic analyses, including spectrogram analysis, single force inversion, and geohazard location. These techniques enable us to determine the physical processes, sliding direction, mass amount estimation, and location of the deep-seated landslide. Through efficient discrete Fourier transform for spectrograms, we identified three distinct events, with the first being the most substantial. Further analysis of spectrograms using a semi-log frequency axis generated by discrete Stockwell transform revealed that Event 1 consisted of four sliding failures occurring within thirty seconds with decreasing sliding mass. Subsequent Events 2 and 3 were minor toppling and rockfalls, respectively. Geohazard location further constrained the source location, indicating that Events 1 and 2 likely originated from the same slope. Subsequently, the sliding direction retrieved from single force inversion and volume estimation was determined to be 153.67° and 557,118 $m^3$, respectively, for the CL. Geological survey data with drone analysis corroborated the above seismological findings, with the sliding direction and source volume estimated to be around 148° and 664,926 $m^3$, respectively, closely aligning with the seismic results. Furthermore, the detailed dynamic process observed in the spectrogram of Event 1 suggested a possible failure mechanism of CL involving advancing, retrogressing, enlarging, or widening. Combining the above mechanism with geomorphological features identified during field surveys, such as the imbrication-like feature in the deposits and the gravitational slope deformation, with event video, infers the failure mechanism of retrogression of the Event 1 after shear-off from the toe. Then, the widening activity was caused by the failure process for subsequent events, as Events 2 and 3. This case study underscores the significance of remote and adjacent seismic stations in offering seismological-based landslide characteristics and a time vision of the physical processes of landslides, thereby assisting geologists in landslide observation and deciphering landslide evolution.

**Keywords:** Cilan Landslide**,** Spectrograms**,** Discrete Stockwell Transform, Landslide Failure Mechanism



## 1 Introduction

In recent decades, seismology has expanded in scope to include mass movements on the Earth's surface since the first observation of landslide signals during volcanic eruptions (Kanamori and Given, 1982). After that, the application of associated analyses from seismology, particularly in landslide research has gradually increased (Brodsky et al., 2003; Vilajosana et al., 2008; Feng, 2011; Allstadt et al., 2013; Hibert et al., 2014; Dietze et al., 2017).

Different seismic signal frequencies play distinct roles in landslide characterization. Low-frequency seismic signals, typically below 0.1 Hz, have been employed to approximate the source location, estimate the sliding direction, and reconstruct its trajectory (Yamada et al., 2013; Hibert et al., 2015; Chao et al., 2018). These signals are generated by ground rebound from slope failure, which were detected in the cases of massive landslides with areas greater than 10,000 $m^2$ or volumes exceeding 100,000 $m^3$ (Kuo et al., 2018). Moreover, low-frequency signals have unique characteristics that can be used as distinguishing features. The signal source is assumed to be a point source that undergoes loading and unloading processes. By comparing synthetic and observed waveforms from a single force mechanism using a grid search and by adapting data from seismic stations, the approximate source location and inverted force direction of a landslide can be determined (Chao et al., 2017). The magnitude of the inverted force is related to the landslide scale (Ekström and Stark, 2013; Chao et al., 2016). However, because of the longer wavelengths associated with low-frequency signals, the accuracy of the source constraints is reduced compared to higher-frequency signals.

High-frequency seismic signals (>1Hz) have different functions in landslide research. They are, for example, used to recognize the details of the source mechanism (Provost et al., 2018; Weng et al., 2022) and the constraints of the source location (Chen et al., 2013; Walsh et al., 2017; Yang et al., 2022). Seismic time-frequency spectrograms have been identified as the source type. Compared with the right triangle spectrogram feature associated with the onset of the P-wave of earthquakes, landslides typically exhibit a cigar-shaped feature with a linear (Suriñach et al., 2005; Moretti et al., 2012) or semi-log (Dammeier et al., 2011) frequency axis resulting from the Fourier transform. More recently, other spectrogram features, such as V-shaped, column-shaped, and pulse-like features corresponding to the





failure mechanisms of sliding, toppling, and rockfall, respectively (Chang et al., 2021) have been observed in spectrograms generated by the Stockwell transform with a semi-log plot. However, the advantages and disadvantages of these two linear and semi-log label transforms have not been thoroughly addressed. Through spectrogram recognition, the duration of the physical processes of a landslide can be determined.

There are three methods used for the source location: (1) time difference (Chen et al., 2013; Fuchs et al., 2018; Manconi and Mondini, 2022), (2) amplitude decay (Aki and Ferrazzini, 2000; Walter et al., 2017), and (3) the azimuth of polarization analysis (Guinau et al., 2019). The time-difference method calculates the time difference between pairs of stations using a velocity model to constrain possible source locations. The accuracy of location determination depends upon the station coverage of the source area (Chang et al., 2023). The amplitude source location method considers the decay of the seismic amplitude with distance. However, the results can be influenced by the distribution of the source-station distances, which often leads to the source location being biased toward the station with the highest amplitude caused by site effect (Chang et al., 2023). Although the first and second methods are commonly used in landslide research, the azimuth of polarization analysis has rarely been discussed in landslide source analysis. Guinau et al. (2019) adapted the polarization to retrieve the source azimuth and locate the rockfall by recognizing P- and S-waves through particle motion.

Investigations into low- and high-frequency seismic signals provide invaluable perspectives on landslides. However, few studies have sorted out the information as preliminary knowledge to geologists, especially for the continuous time vision of the failure process. Historically, geologists relied solely on field and drone surveys conducted before and after landslide events to depict landslide failure mechanisms, thereby lacking associated information on temporal evolution to link different phases of landslide activity. Geologists need to speculate on the connection of landslide activity to the geological model. However, seismological-based information complements this approach by providing temporal context. Therefore, this study integrates seismic results with landslide investigation (field and drone surveys) to illustrate constructing a landslide evolution model.





## 2 Background information

### 2.1 Landslides During Typhoon Nesat

Torrential rainfall resulting from the interaction between Typhoon Nesat and the winter monsoon lashed Yilan County in northeast Taiwan from 15 October to 17 October 2022 (Fig. 1a; the time in this research all shows in local time UTC+8). The accumulated precipitation reached 1,000 mm in three days, with a peak rainfall intensity of 103 mm hr$^{-1}$. (Fig. 1b). This accumulation exceeded the landslide threshold of 550 mm, as documented by the Agency of Rural Development and Soil and Water Conservation (https://246.ardswc.gov.tw/; last accessed on 4 April 2024). Consequently, many rainfall-induced landslides occurred, destroying sections of two vital provincial highways: No. 7 (Northern Cross-Island Highway) and No. 7A (Fig. 1a). The Directorate General of Highways, Taiwan, reported nine sections damaged by landslides (Figs. 1c-1j; Table S1). Among these, three roadbed washouts characterized by argillite/slate were observed in the Paling Formation (Figs. 1c-1e), whereas a landslide composed of slate covered a road section in the Lushan Formation (Fig. 1f). Additionally, four debris flow events occurred near the boundary between the Lushan Formation and the alluvium (Figs. 1g-1j). As of 22 October 2022, these events left 302 people stranded, resulting in one missing person. In addition to these nine events, a deep-seated landslide, known as the Cilan landslide (CL), occurred in the Lushan Formation. Initially, an individual captured a part of the CL process on video, revealing two distinct stages of material sliding (Fig. S1). According to the video footage, the initial landslide was formed by exposure to the bare earth. The video captured the subsequent failures. The first body in the footage slid between 0-20 seconds, and the following failure occurred at the 20-second mark in the video recording. Then, the Directorate General of Highways in Taiwan identified the precise location of the landslide (Fig. S2 and yellow star in Fig. 1a) and provided an approximate occurrence time of 4:00 PM on 16 October 2022.







Figure 1 (a) Regional geologic map of roadside landslides, seismic station, and rain gauge (Fei and Chen, 2013). The grey-shaded area in the map of Taiwan (lower left) represents Yilan County. (b) Rainfall data of rain gauge 01U060 during the typhoon. The rain episode started on 15 October 2022 at 00:00 AM (Local time UTC+8). (c)-(j) In situ photos for landslides. All photos are open data from the Directorate General of Highways, Taiwan (Table S1). The k in the map/photos indicates the milestone (in kilometers) of two provincial highways.



## 2.2 Topographic Feature Near the CL


According to 1-m high-resolution LiDAR-derived slope inclination map (Fig. 2), several
prominent features were evident near the CL. On the west side of the CL, a concave slope displayed
distinct scarps and gullies. These features strongly indicated that the concave slope was subject to erosion.
Conversely, the eastern roadside slopes of the CL revealed a contrasting topography characterized by
numerous scarplets with several gullies and convex slopes, all prone to rockfalls. Within this context,
slopes at the 86.5 km and 86.7 km milestones along Provincial Highway No. 7 experienced a debris flow
on the gentler portions (Fig. 1h) and talus deposits on the steeper sections (Fig. 1i).

Figure 2 Topographic feature interpretation with 1 m high-resolution LiDAR slope map in 2014 before
the CL. The AA' is for the topographic profile in Fig. 7b. The black line polygons indicate the
source boundary of slope failures.





## 3. Methods


The study aims to leverage seismic analysis as preliminary knowledge to aid in illustrating the
landslide evolution model of the CL. To achieve this, we conducted a series of seismic signal analyses,
including seismic signal spectrograms, single force inversion (SF), and geohazard location (GeoLoc).
These analyses provide insights into the temporal evolution of the failure process, landslide magnitude,
inverted force direction, and landslide location constraints. Geologists use these results to gain a basic
understanding of the CL prior to conducting a field survey. The field survey consists of two parts:
geological investigation and drone survey. The former provides geological background information,
while the latter joining LiDAR in 2014 captures topographic features and changes post-CL. The combined
results from the seismic signal analyses and field surveys support the development of the most plausible
landslide evolution model. The flowchart is depicted in Fig. 3.

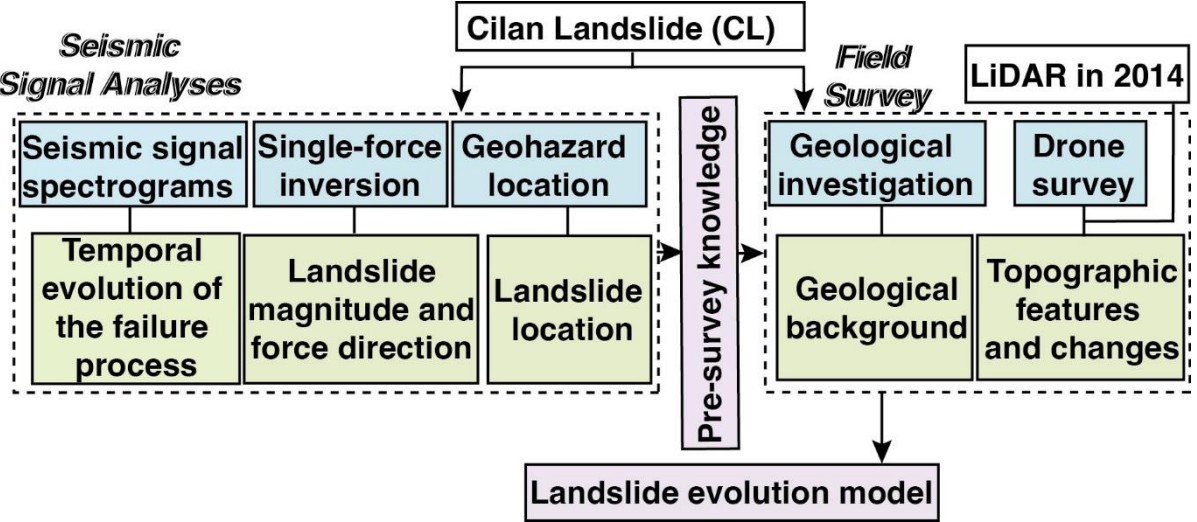


Figure 3 Flowchart of this study. The blue, green, and purple backgrounds are relevant to the methodology,
results, and discussion.



## 3.1 Seismic signal spectrograms

The study investigated time-frequency spectrograms based on the power spectral density (PSD) of the discrete Fourier transform (DFT), as well as the power spectrum (PS) of the discrete Stockwell transform (DST). The DST of Eq. (1) was derived from Eq. (2), while $f$, $\tau$, t, and α were derived from Eqs. (3)-(7).

$$DST:\ s[p\Delta t, \tfrac{l}{N\Delta t}] = \sum_{m=1}^{N-1}\ H[\tfrac{l+m}{N\Delta t}]e^{-2\pi^2\frac{m^2}{l^2}}e^{2\pi i\frac{mp}{N}} \tag{1}$$

$$DFT:\ H[\tfrac{l}{N\Delta t}] = \tfrac{1}{N}\sum_{k=0}^{N-1}\ h[k\Delta t]e^{-2\pi i\frac{lk}{N}} \tag{2}$$

$$f = \tfrac{l}{N\Delta t}\ ,\ l = 0,1,2\ldots N-1 \tag{3}$$

$$\tau = p\Delta t\ ,\ p = 0,1,2\ldots N-1 \tag{4}$$

$$\alpha = \tfrac{m}{N\Delta t}\ ,\ m = 0,1,2\ldots N-1 \tag{5}$$

$$t = k\Delta t\ ,\ k = 0,1,2\ldots N-1 \tag{6}$$

$$PS = s[p\Delta t, \tfrac{l}{N\Delta t}]^2 \tag{7}$$

where $\Delta t$ is the time sample interval, $\tau$ denotes the time of spectral localization, $N$ is the total number of data points, α, and $f$ control the discrete frequency point, and $h[t]$ is the discrete-time series seismic data.

In the context of landslides, the predominant frequencies of ground vibrations typically range from 1 Hz to 10 Hz (Chang et al., 2021). To represent the power distribution within this range precisely, we configured the DFT analysis to have time and frequency resolutions of 1.28 seconds and 0.39 Hz, respectively. This configuration effectively captured the pertinent frequency information while retaining an acceptable time resolution. Also, a cumulative PSD plot was obtained by summing the PSD values at discrete time intervals. Alternately, applying DST instead of DFT allows for either enhanced frequency resolution for the lower frequencies through broader time windows or improved time resolution for the higher frequencies through narrower windows. In this study, we opted for a time window of 0.05 seconds and a frequency resolution of 0.30 Hz in the DST analysis. These parameters provided superior frequency and time resolutions, enabling the capture of intricate spectrogram details.




The scale of the frequency axis on the spectrograms profoundly influences recognition and interpretation within the target frequency range of 1 Hz to 10 Hz. Therefore, we incorporated linear and logarithmic frequency axes into the spectrograms. By judiciously selecting window lengths, time and frequency resolutions, and frequency axes, we facilitated effective visualization and analysis of the power distribution in seismic signals, particularly within the frequency range pertinent to landslide occurrences.

**3.2 Single-force inversion (SF)**

Single-force inversion (SF) is a technique used in the near-real-time landquake monitoring system (NRLAMS) (Chao et al., 2017) to extract the possible force direction and magnitude of a landslide. Before conducting the SF analysis, we performed several preprocessing steps on the seismic signals. First, we applied a bandpass filter between 0.02 Hz and 0.05 Hz to isolate the frequency range for large-scale landslides (volume $> 10^5$ m$^3$ or area $> 10^4$ m$^2$, as defined by Chen, 2015). This frequency range is associated with landslide-related signals in Taiwan (Chao et al., 2017). In addition, we transformed the original horizontal components of the seismic data into radial and tangential components. Different weightings in the SF correspond to the signal-to-noise ratio (SNR) (Table S2), the ratio between the absolute peak amplitude and the average absolute amplitude from the entire signal trace.

Subsequently, the SF analysis simulated synthetic waveforms assuming a source depth of 1 km, and Green's functions were calculated based on the surface wave velocity model proposed by Shin and Chen (1998). Different synthetic waveforms were generated using different settings of force direction, magnitudes, and dips. These waveforms were compared with the observed signals regarding fitness values, the sum of the maximum normalized cross-correlation coefficient, and variance reduction. The highest fitness values corresponding to the inverted force parameters were determined. Furthermore, a parameter of inverted force magnitude (unit: Newton) of SF could be used to estimate the landslide mass through the empirical formula, mass (kg) = 0.405 × force magnitude (Chao et al., 2016). Assuming a rock density of approximately 2,600 kg m$^{-3}$, the estimated landslide mass could be roughly converted to landslide volume. The seismic data for the SF analysis were obtained from a broadband array in Taiwan for seismic networks (Kao et al., 1998). A more detailed methodology associated with the parameter setting and procedure is provided by Chao et al. (2017).



## 3.3 Geohazard location

The geohazard location (GeoLoc) method, as outlined by Chang et al. (2021), synergizes the cross-correlation (CC) method (Chen et al., 2013) with the amplitude source location (ASL) method (Aki and Ferrazzin, 2000) to pinpoint potential landslide locations using seismic signals in the frequency range of over 1 Hz. This approach initially filtered the seismic data between 1 Hz and 3 Hz. Subsequently, the SNR was calculated as a ratio between the short-term average (± 5 s from the maximum envelope amplitude) and the long-term average of a 180-second target trace. A threshold of SNR larger than 1.7 was applied to select the available waveforms for further analysis. The selected frequency range and SNR threshold were empirically established based on extensive-scale landslides in Taiwan (Chen et al., 2013).

The CC method calculates the maximum cross-correlation coefficient between each station pair to extract the travel time difference. This difference was then used with a three-dimensional velocity model (Wu et al., 2007) and grid search to define the misfit function. Simultaneously, the ASL method gauges its misfit function by optimizing the fit of the amplitude decay curve. By individually sorting the misfit functions across all search grids, both methodologies yielded reliable source locations for landslides. The potential source locations were identified when the grids had relative fitness values greater than 0.95 Chang et al. (2023). The detailed algorithm of GeoLoc can be found in Chang et al. (2021).

Seismic data for GeoLoc analysis were collected from various sources (Table S3), including temporary stations maintained by the Comprehensive Landquake Monitoring Lab (CoLLab), National Yang Ming Chiao Tung University, a broadband array in Taiwan for seismology, and the Central Weather Administration, Taiwan.

## 3.4 Field survey

The field survey encompassed two integral components: drone survey and geological investigation. For the drone survey, we conducted a series of vertical and inclined aerial photos along the CL using the DJI Phantom 4. These photos were input for the photogrammetry software Pix4D, which generated a digital surface model (DSM). Through the DSM, the geomorphological features after the CL could be depicted. Additionally, by combining the LiDAR data from 2014 with DSM, we could observe





the differences in topography before and after the CL event. On the other hand, the geological
investigation focused on road inspection and outcrop observation before and after the CL, respectively.
The road inspection documented the status of slope protection, particularly regarding crack geometry.
The outcrop observation recorded the strike and dip for cleavage, joints, and bedding near the CL.

**4. Result**

**4.1 Seismic spectrograms for the CL**

According to the one-hour spectrogram generated through DFT analysis of V03G (Fig. 4a), the
resulting spectrograms revealed three distinct high-power onsets potentially corresponding to different
landslide events. The first event, labeled Event 1, occurred from 16:10:00 to 16:10:30. The cigar-shaped
features of the spectrograms indicated a landslide process when the spectrograms were analyzed using a
linear frequency axis (Fig. 4b). However, when the DST was modified to a semi-log graph, the lower
bounds of the high PSD displayed V-shaped spectrogram features (Fig. 4c). Such V-shaped patterns were
not discernible in the spectrograms obtained using the DFT because of the inherent limitations imposed
by the frequency and time resolution (Fig. 4b). The V-shape is associated with sliding behavior (Chang
et al., 2021), which involves phases of acceleration and deceleration of the landslide materials separated
by the lowest point of the V-shape. In Event 1, the four V-shaped events were interconnected, and their
lowest points (depicted as purple dots in Fig. 4c) gradually shifted to higher frequencies, indicating a
reduction in the sliding material volume. As a result, the most significant event was the initial sliding of
massive mass, which also generated signals within the frequency range of 0.02 Hz to 0.05 Hz, detectable
by the SF method (See following subsection). With smaller volumes involved in the subsequent sliding
events, the corresponding signals in the low-frequency range could not be generated.
Approximately 20 min after Event 1, Event 2 occurred, and the spectrogram revealed a sequence
of continuous pulse-like features (black rectangle in Fig. 4d). However, the frequency bands associated
with these pulse-like features overlapped with ambient noise. Certain pulse-like features could be
discerned, indicating continuous rock-ground impacts in the form of rockfalls. Subsequently, a column-
like shape emerged in the spectrogram, an interaction between the substantial mass and the slope or
ground (dashed black rectangle in Fig. 4d). This phenomenon referred to processes such as toppling or



rockfalls on overhanging slopes or similar mechanisms. Approximately 26 min later, Event 3 emerged,
presenting spectrogram features analogous to the continuous rock-ground impacts observed in Event 2
(black rectangle in Fig. 4e). In particular, a gradual decrease in the PSD and PS values and signal durations
was evident from Events 1 to 3, presenting a reduction in the scale of the landslide.

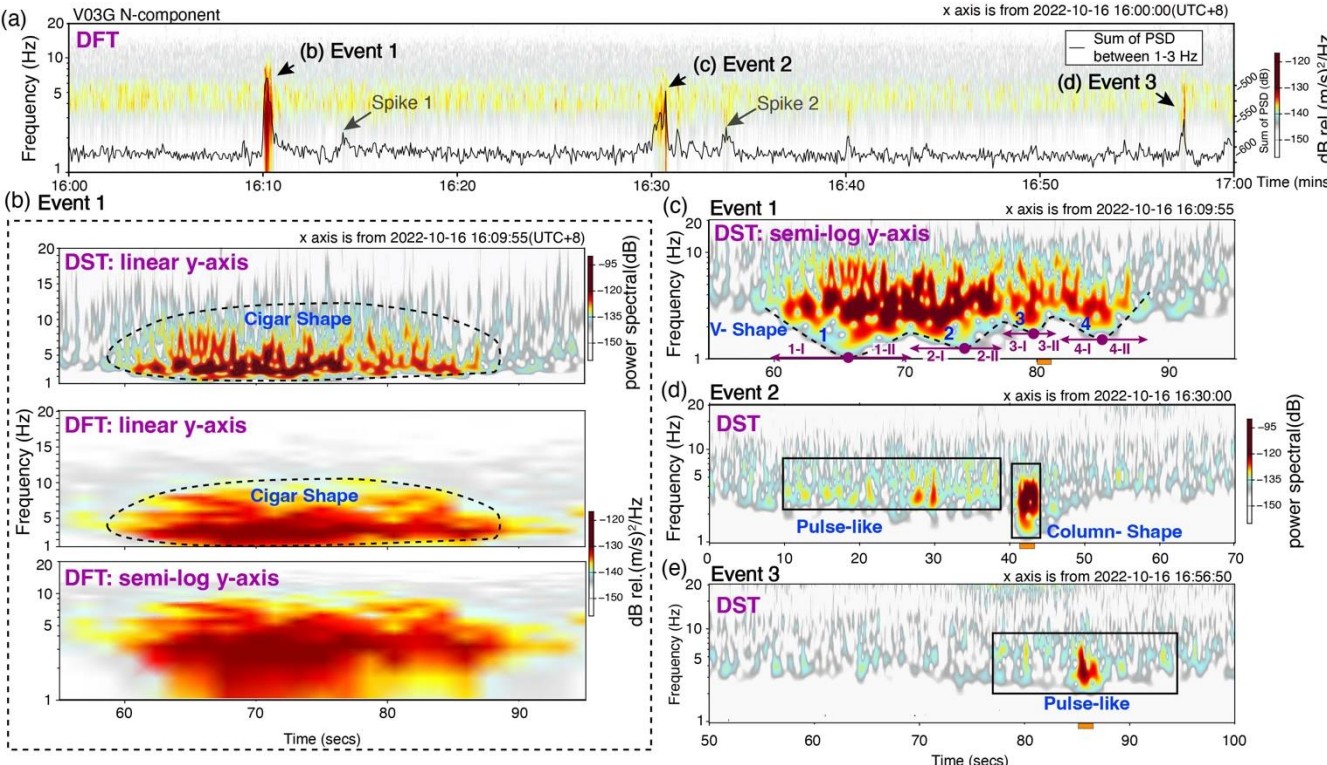


Figure 4 Spectrograms of the seismic signal from V03G with the North component. (a) DFT spectrogram
and PSD sum between 16:00-17:00 on 16 October 2022 (UTC+8). (b) Event 1 of DST with linear
frequency axis and DFT with linear frequency and semi-log frequency axes. (c) DST spectrogram with
semi-log frequency axes for Event 1. The black dashed line is the lower boundary of the high PS values,
showing the V-shaped spectrogram feature. The purple points are the lowest points of the four V-shapes
that separate the first half (I), the acceleration phase, from the second half (II), the deceleration phase.
The blue numbers indicate a sequence of the V-shapes. (d) Spectrogram of DST for Event 2. The black
rectangles mark the spectrogram feature. (e) DST spectrogram for Event 3. The horizontal orange bars
below the x-axis in (c)(d)(e) are the signal windows for particle motion analysis in Fig. 10.





Except for Events 1 to 3, Fig. 4a exhibited two spikes. We examined the corresponding
spectrograms and found that the signals were faint and heavily obscured by ambient noise (Fig. S3).
Consequently, the evidence derived from these indirect observations does not substantiate their origin
from landslide activity.

**4.2 Single-force inversion for the CL**

We employed an SF approach for Event 1 of the CL, utilizing a network of five seismic stations
(Fig. 5a). Source-station distances spanned from 7.80 km to 124.13 km, and back azimuths ranged from
170º to 296º. After testing several starting times of the seismic signals for the SF, we found that signals
starting at 16:10:04 yielded the best results. The normalized cross-correlation coefficient and the variance
reduction of these signals averaged 0.72 and 0.74, respectively (Fig. 5b). The overall performance
exhibited a fitness value 1.08. Subsequently, the SNR values ranged from 2.93 to 6.00, and the NACB
consistently exhibited a relatively high SNR across the three components. The inversion process yielded
a force direction of 153.67° and a force magnitude of $3.36 \times 10^9$ (Newton). The magnitude of the force was
converted into landslide mass using an empirical formula, and the landslide volume was estimated to be
approximately 523,540 $m^3$.





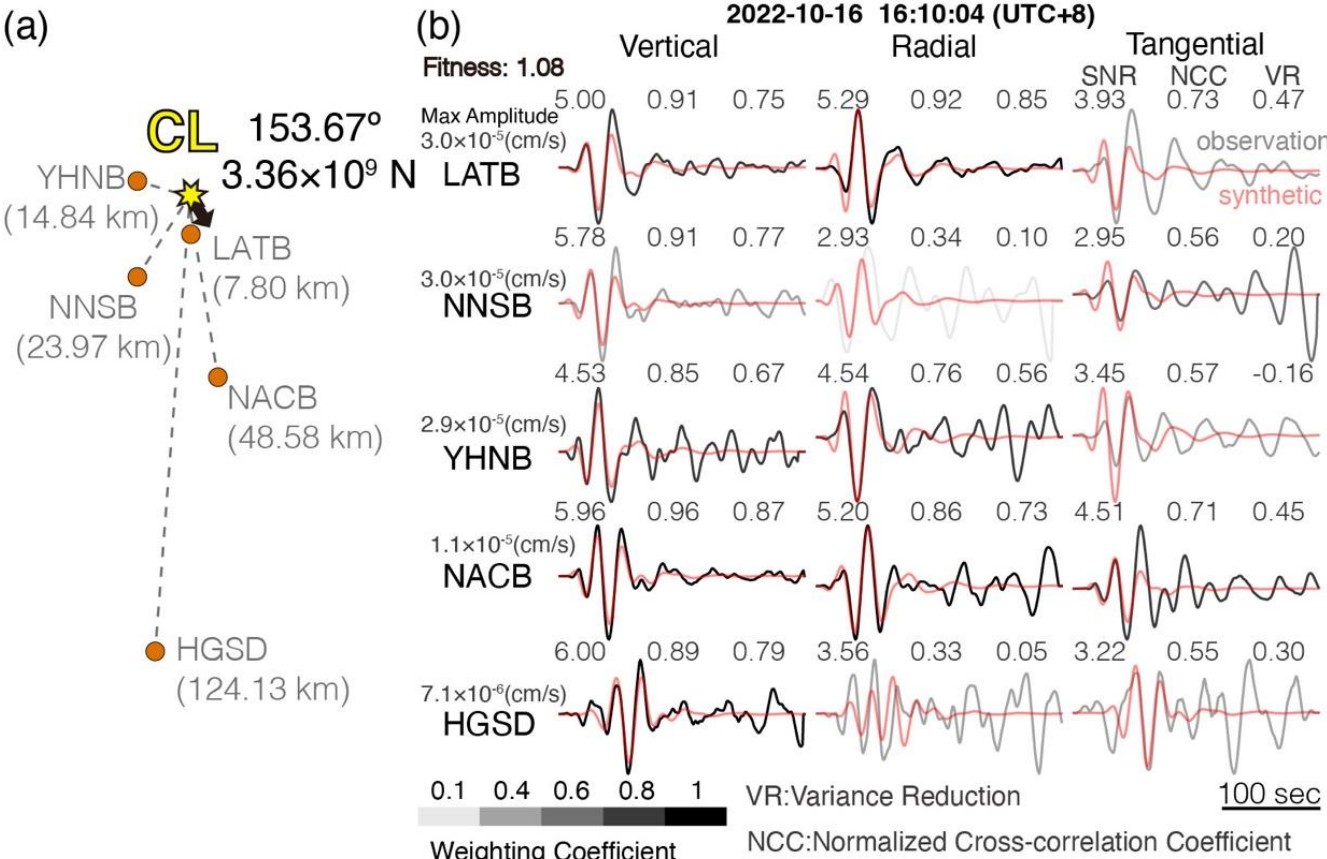

Figure 5 (a) Position of seismic stations relative to the CL. The black arrow indicates the inverted force
direction of 153.67° with a force magnitude of 3.36×10⁹ (Newton). (b) Synthetic and observation
waveforms of the CL with SNR, normalized cross-correlation coefficient, and variance reduction. The
grey gradient presents the different weightings to retrieve the fitness corresponding to the SNR of the
signals (Table S2).

**4.3 Source location**

The V03G station recorded Events 1 to 3. When these signals originated from the same slope of

landslides, the characteristics of the spectrogram could provide valuable insights into the short-term

behavior of the CL. Therefore, to determine the locations of Events 1 to 3, we utilized the GeoLoc method.

The results of Events 1 and 2 of the CC, considering both the horizontal and vertical components,

indicated that the grids with high fitness values (> 0.95) were close to the V03G station (Purple grid cells

in Fig. 6). In addition, the ASL tended to be near the station with the highest amplitude (Fig. S4).





Therefore, Events 1 and 2 probably originated from the same landslide location. For Event 3, the signals
were too weak to be detected by the ENT and LATB stations. Only V03G recorded its signals, which
supports the location of Event 3 near the V03G station.

Figure 6 Location determination by cross-correlation-based method of horizontal and vertical component
data for Event 1 and Event 2. The values following the stations with transparent white backgrounds are
SNR.





## 4.4 Landslide survey

According to the topographic profile (A-A' in Fig. 7a), the sliding direction was approximately 148°, similar to the result obtained from the SF (153.67°). The observed elevation difference and travel distance of the CL were 220 m and 530 m, respectively. The apparent friction angle ranged from 22° to 31°, transitioning from the main scarp to the first significant and subsequent failures (Fig. 7b). This variation could be attributed to the depositional environments and landslide volumes.

Furthermore, we compared the topographical profiles before (1m high-resolution LiDAR data in 2014) and after (DSM by drone) the CL. The data revealed that the maximum erosion depth approached approximately 45 m near the left flank of the CL, where the bedrock was exposed. For the location, photographic evidence shows that the dips of slate cleavage exhibited a gradual transition from steep (at the top) to gentle (at the bottom) (Fig. 7b). This characteristic indicated gravitational slope deformation (Chigira, 1992; Agliardi et al., 2001), suggesting a weakening of the structural integrity and strength of the rock mass constituting the slope. The CL originated from a source area measuring 44,562 m$^2$ and was deposited over an area of 94,396 m$^2$, resulting in a maximum colluvium thickness of 30 m (Fig. 7b). The calculated source volume by difference of elevation was approximately 664,926 m$^3$. Consequently, the landslide mass was converted into a deposited volume of 690,445 m$^3$.

The slope map of DSM exhibited deposits with imbrication-like features at the landslide toe, which was covered on the wider and flatter colluvium with the first toe. This pattern was contributed to by the widespread colluvium area where numerous trees rest on the colluvium, composed of slate boulders, debris, saprolites, and soils (Figs. 7b and 8). The inclined trees on the colluvium imply the colluvium is displaced with slight disturbance due to the low-friction basal detachment. The imbricated deposits near the original roadside slope represented a depositional sequence resulting from later failures. The result of the geological investigation shows that the dip direction of the slate cleavage corresponded to the slope aspect and sliding direction, with a high dip angle influencing CL failure (Fig. 7b). Additionally, before the occurrence of the CL, an inspection conducted in May 2022 revealed slight damage and displacement of the downslope concrete wall near the slope (Figs. 9a-9e). These damage signs served as early indicators of creeping slope.





Figure 7 (a) Topographic feature interpretation on drone-based slope map after landslide (19 October 2022). (b) Topographic profile of AA'. The embedded drone photos show the slate outcrop on the left flank of the landslide. The dashed curves indicate that the dip angle and traces of cleavages changed and deformed.



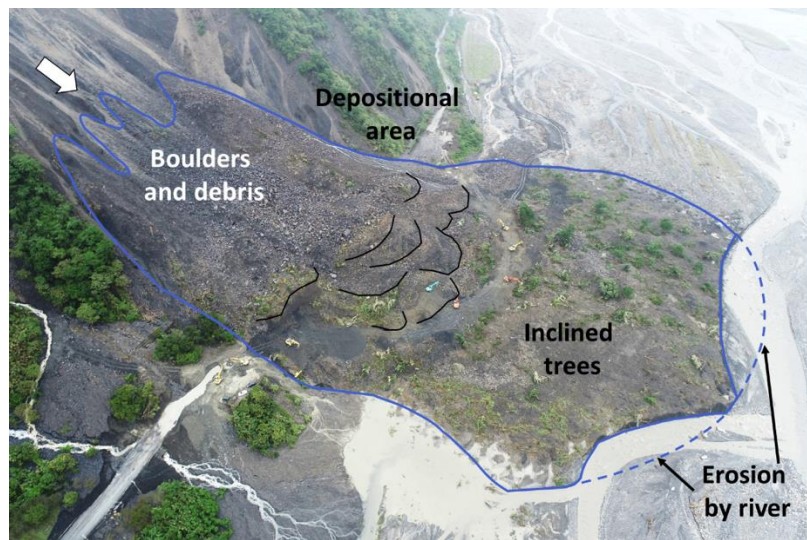

Figure 8 The closed aerial photo of CL deposits. The imbrication-like features are lineated by black lines. The first toe was eroded by river.

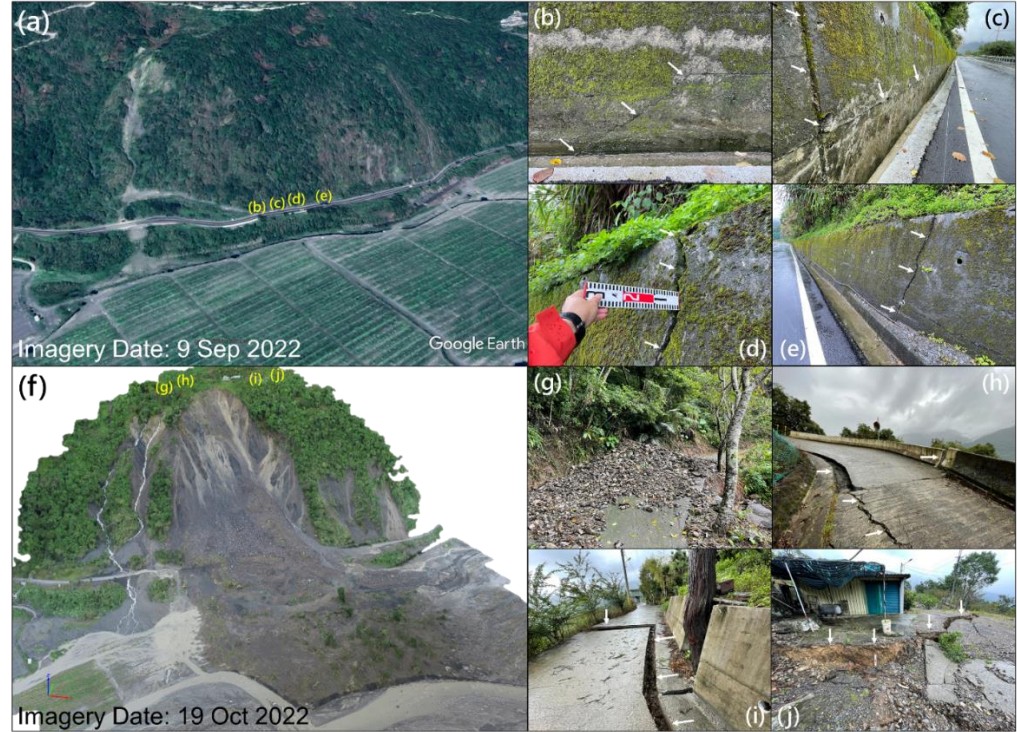

Figure 9 (a) © Google Earth image before the CL. (b)-(e) Photos from inspection on 27 May 2022. (f) Drone-based 3-dimensional model after the CL. (g)-(j) Photos of crown cracks of the CL.





## 5. Discussion

### 5.1 Source location of landslide signals

Assuming that Event 1 to Event 3 originated from the slope of the CL, we performed a particle motion analysis within 1 Hz to 3 Hz of V03G (back azimuth: 208°) to clarify their source locations. Given the surface behavior of the landslides, our analysis focused on the vertical and radial components associated with the propagation of Rayleigh waves. Regarding Event 1, we observed intricate patterns of particle motion, particularly during phase 1 of the sliding (Fig. S5). While the initial sliding phase of Event 1 indicated significant movement, the event was governed by a single force mechanism. However, not all materials involved in the landslide exhibited pure shear sliding. Some materials were bouncing, rolling, or interacting with the ground, slopes, and adjacent particles. These physical processes could generate high-frequency signals, resulting in complex and inexplicable particle motion patterns from phase 1 to phase 2 of Event 1. In contrast, the particle motion displayed a more consistent direction during the small-scale mass movement observed in phases 3 and 4, which manifested as clear ellipses. Notably, the ellipse corresponding to phase 3-II of Event 1 was particularly pronounced and linked to Events 2 and 3 (Fig. 10). These elliptical patterns indicated retrograde motion along the travel direction.

Further, we investigated the relationship between different back azimuths and elliptical shapes. When the back azimuth was set at 228°, the elliptical shapes showed a noticeable change (Fig. S6a). However, similar elliptical shapes were consistently observed for back azimuth values ranging from 188° to 148° (Figs. S6b-c). This suggests that elliptical shapes do not exhibit significant sensitivity within 208° to 148°, where sources could potentially originate from a wide back azimuth range. Nevertheless, no documented landslides are reported during the range of back azimuth between 208° to 148° (Fig. S7); most substantial landslides are oriented with back azimuth angles greater than 228° (Fig. 2). Subsequently, the CC and ASL results indicated that Event 2 is close to V03G. Therefore, we posit that at least Event 1 and Event 2 likely originated from the same source direction. Event 3, the minor event, challenges determining its precise location by seismic analysis.



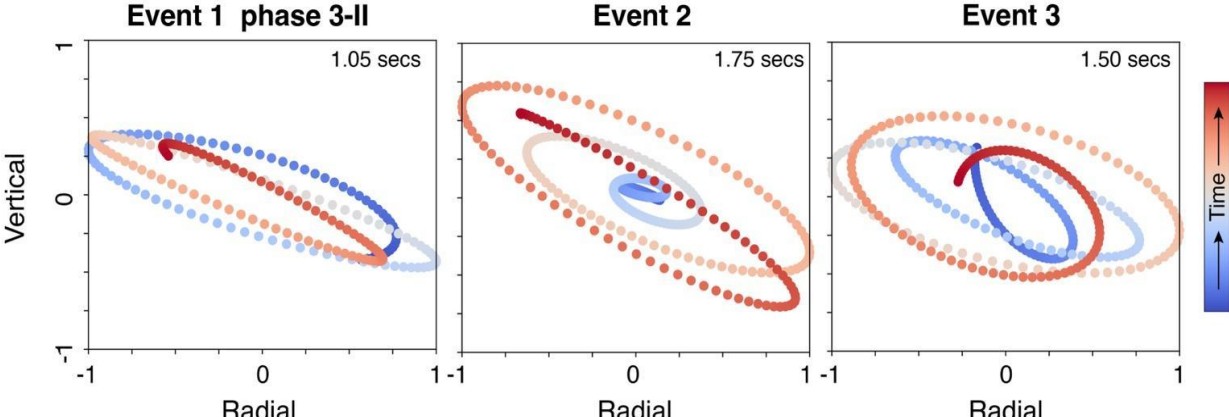

Figure 10 Particle motion comparison between radial and vertical components for phase 3-II of Event 1, Event 2, and Event 3. The duration of particle motion signals is indicated in the upper right corner of each subfigure, with precise timing details highlighted by orange bars in Fig.4.

## 5.2 Comprehensive information from seismic analysis

When Event 1 and Event 2 occurred on the same slope, we estimated the approximate volume of the CL using empirical regressions. Event 1 indicated a volume of 523,540 m$^3$ based on empirical regression (mass = 0.405 × force magnitude; Chao et al., 2016) for large-scale landslides with a sliding direction of 153.67° (Table 1). By retrieving the amplitude at the source (A$_0$; unit: cm s$^{-1}$) through 1-8 Hz of horizontal signals, we estimated the volume of Event 2 is 16,791 m$^3$ (Volume= 77,290 A$_0^{0.44}$; Chang et al., 2021). The total volume obtained from a seismic analysis output of 540,331 m$^3$ is around 19% lower than the volume estimated by the difference between LiDAR and DSM.

Seismic signal analysis provides valuable insight into potential landslide processes. The SF method detailed the timing and movement direction. The DST spectrogram analysis revealed distinct timeframes and physical patterns of the three events associated with slope failure. Event 1 likely involved four sliding failures within 30 seconds with gradually decreasing masses. However, the landslide video captured the process, lasting approximately 34 seconds for probable phases 2 and 3 in Event 1. This discrepancy in timing may be attributed to the weak kinetic energy during the early and termination stages of the CL. The ground vibration signal at those stages might not transmit to V03G, potentially influencing





the recorded duration. Subsequently, Event 2 featured 30s of continuous rockfall, followed by a toppling
event with a larger mass.

**Table 1** Pre-survey information of the CL by seismic analysis

| Landslide characteristics | Information | |
|---|---|---|
| | Event 1 | Event 2 |
| Occurrence time | 4:10 PM on 16 October 2022. | 4:30 PM on 16 October 2022 |
| Estimated volume | 523,540 m$^3$ | 16,791 m$^3$ |
| Sliding direction | 153.67° | - |
| Failure process | Four continuous sliding with the gradual reduction in sliding volume | Rockfall and toppling |


**5.3 Landslide evolution model**

Landslides are categorized into seven movement types (Varnes, 1978). According to the

spectrogram features of Event 1 in the CL, this seismic analysis preferred sliding movement. The stepwise
failure process of a landslide can be determined based on the distribution of landslide activity, such as
advancing, retrogressive, enlarging, or widening activities (Fig. 11) (WP/WLI, 1993). Advancing and
retrogressive activity involves the expansion of the rupture surface along and in the opposite direction of
movement. Enlargement entails the rupture surface expanding in multiple directions, whereas widening
indicates that the rupture surface extends into one or both flanks of the landslide.



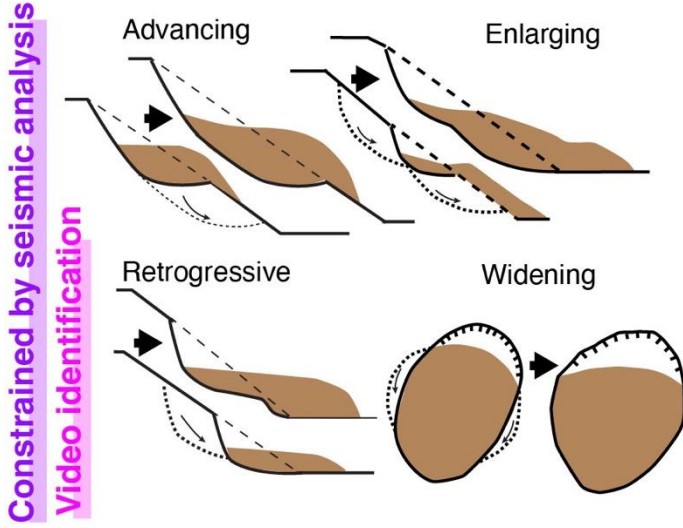


Figure 11 Schematic diagram of potential landslide activities for the CL. Dot lines represent the detachments in the next stage. Dash lines indicate the original ground level. The brown color indicates the extent of displaced material. The figure has been modified from Cooper (2007).

For the CL, the left flank was exposed at the beginning of the video (Fig. S1), which implies previous failures had occurred earlier than the beginning of the video. It is reasonable that people were attracted by previous failures and prepared to take videos for subsequent failures. Besides, the debris flow had deposited a debris fan on the road (Fig. 2h), and excavators were operated to clear the buried road section. The moving mass from the upslope was sliding downward and pushing the previous colluvium. Moreover, some trees displaced to downslope on the top of moving mass, which implies a similar phenomenon during early failure with similar landslide source conditions (Fig. 7). Consequently, the video recorded the process of sliding that could correspond to the distribution and geometry of deposits (Figs. 7 and 8) and validate the sliding of Event 1 by seismic analysis (Fig. 4d).

Combining the pre-survey understanding with the survey results, the initial model for landslide evolution was established (Fig. 12a). According to the announces of the Directorate General of Highways in Taiwan (Table S1) and the CL occurrence time extracted by seismic analysis, the debris flow and rockfall were induced by heavy rainfall before the CL (Fig. 12b). The initial failure mechanism of the CL could be assumed to initiate at a shear-off from the original toe (roadside slope) caused by the high pore-water pressure after heavy rainfall infiltration. The first failure should deposit on sandy and gravelly




alluvial deposits with high water levels (Fig. 12b). The rapid loading of sliding mass onto the wet alluvial

deposits may have induced liquefaction and reduced the basal friction of sliding mass (Sassa, 1992). The

first failure had the most significant volume, leading to higher mobility (Figs. 7, 8, and 12c) (Corominus,

1996; Legros, 2002; Hungr and Evans, 2004). It possibly corresponds to Phase 1 of Event 1 in Table S1.

However, for subsequent failures (Phases 2-4 of Event 1), failure masses were deposited on the angular

debris and boulders of the previous colluvium, characterized by a rough ground surface, resulting in lower

mobility (Fig. 12d). This process of retrogression may be captured in the video (Fig. S1). Therefore, the

most plausible landslide activities could be retrogression in Event 1. Then, the widening activity

developed by toppling and rockfalls with subsequent Event 2, Event 3, and other failures from the steep

scarp and flanks (Fig. 11) (refers to YouTube video: https://www.youtube.com/watch?v=PMlb7OiCqMQ;

last access on 2 April 2024).

The continuous presence of the four phases in Event 1 (Fig. 4b) and the field survey imply a

fractured bedrock/steep sliding surface near the CL. Following the landslide, steep scarps and flanks

emerged, exhibiting discontinuities such as cleavage, joints, and numerous tension cracks at the crown

(Figs. 9f-9j). This observation suggests the potential for further enlargement of the landslide. An unstable

slope directly threatened the safety of residents living close to the crown of the CL. Therefore, it is

imperative to implement comprehensive monitoring measures on the slopes described by Kang et al.

(2021). These measures are essential for gaining a deeper understanding of ongoing landslide activity and

ensuring the safety of the affected population.



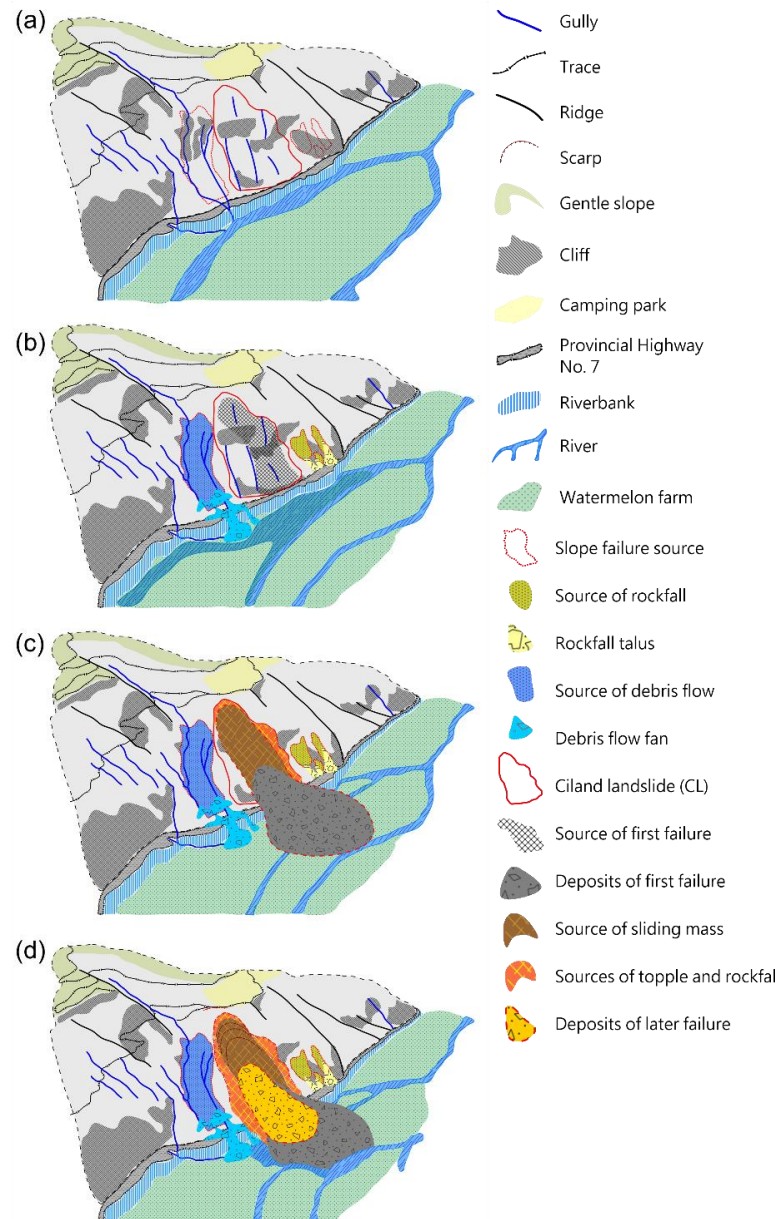

Figure 12 The geological model and topographic evolution of the CL. (a) The initial model is based on LiDAR topographic features. (b) According to Fig. 2 and Table S1, slope failures (debris flow and rockfalls) occurred around the CL. (c) The failure process of Phase 1 of Event 1 of the CL is based on Figs. 4 and 7 and Table 1. (d) The final stage of the CL, after Phases 2-4 of Event 1, Events 2 and 3 of the CL.





## 6. Conclusions

Research on the Cilan Landslide (CL) has shown how to deliver seismic analysis results as pre-survey knowledge to geologists for field surveys. We investigated a series of events involving the efficient generation of one-hour spectrograms through a discrete Fourier transform. Three events, Event 1, Event 2, and Event 3, were identified, with four continuous phases of sliding, rockfall, and the subsequent toppling and rockfalls, as revealed by the spectrograms obtained via a Stockwell transform with a semi-log frequency axis. The initial sliding of the CL generated low-frequency seismic signals (ranging from 0.02 Hz to 0.05 Hz), and we successfully determined an inverted single force direction of 153.67º, close to the actual direction of landslide movement, which was 148º. This geohazard location (GeoLoc) pinpointed Event 1 and Event 2 close to seismic station V03G, whereas a polarization analysis provided further support for the notion that these event sources may have originated from the same direction, indicating a high probability of sharing the same slope of origin. Additionally, by employing the force magnitude and amplitude at the source ($A_0$) in the empirical regressions for Event 1 and Event 2 of the CL, we estimated the landslide volume to be 540,331 $m^3$, 19% lower than the volume calculated using a digital elevation model.

This information has significantly contributed to geologists' understanding of the physical processes underlying the CL for predicting advancing, retrogressive, enlarging, or widening mechanisms. After combining the field survey, the seismic analysis results have led geologists to propose a detailed mechanism for the CL. This mechanism involves shear-off from the roadside slope and subsequent mass sliding triggered by high pore-water pressure from rainfall infiltration. The observed physical behaviors of subsequent failures and topographic features with imbrication-like deposits suggest that the most plausible landslide activity may undergo retrogression and widening over time.

The research supported the idea that seismic analysis enables the determination of a landslide's inverted-force direction, estimated landslide volume, and physical processes. Notably, seismic analysis from an adjacent station provides additional temporal insight into landslides' dynamics, whereas geological surveys can only investigate the topography post-landslide to constrain the failure mechanisms. Therefore, seismic analysis provides crucial information for geologists before conducting field surveys.



**Data availability**

Waveform data for this study were provided by the Broadband Array in Taiwan for Seismology (BATS; https://doi.org/10.7914/SN/TW, Academia Sinica, Institute of Earth Sciences, 1996.) and the Central Weather Administration, Taiwan (CWA; https://doi.org/10.7914/SN/T5). The raw seismic data of V03G is available through Figshare (https://doi.org/10.6084/m9.figshare.24464281.v1). The digital terrain model (DTM) of the 20-meter resolution used in Fig. 2 is available from the Government Open Data Platform, Taiwan (https://data.gov.tw/dataset/35430; Ministry of the Interior, 2024). The road shape files are available from the National Land Surveying and Mapping Center, Taiwan (http://maps.nlsc.gov.tw/S_Maps/wms). The last accessed of all URLs was on 4 April 2024.

**Competing interests**

The contact author has declared that none of the authors has competing interests.

**Author contribution**

JM and CM conceived of the presented idea. WA supervised the project and provided critical feedback and helped shape the research, analysis and manuscript. JM, CM, WA, MW carried out the field investigations. CS, MW, TC, and CY discussed the results and contributed to the final manuscript.

**Acknowledgments**

The authors acknowledged the National Science and Technology Council of Taiwan (NSTC) for the funding support. The authors acknowledge the Forestry and Nature Conservation Agency, Taiwan for providing LiDAR data and the Geological Survey and Mining Management Agency, Taiwan for LiDAR data establishment.

**Financial support**

This study is financially supported by the National Science and Technology Council of Taiwan (NSTC) for Che-Ming Yang under grants NSTC 110-2116-M-239-001-MY2, NSTC- 112-2116-M-239-001, NUU project No. SM113004 and We-An Chao under grants NSTC 111-2625-M-A49-004-MY3.



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
