# Peer review of "Unravelling Landslide Failure Mechanisms with Seismic Signal"

_EGUsphere, 2024_

## Author Comment (AC1)

**Responses to Reviewer's Comments**

**Note**: Reviewer's comments are all quoted in their entirety and are in *Italics*, while authors' responses are in blue.

**# Reviewer 1**

I found your paper very interesting, especially from the fact that you can use seismology data to recreate the slope movement and direction. I cannot say I have anything to add to the paper.

The only one thing I would like to address is the use of the method for the landslide size and type. From the data presented in the paper, it looks like the method is completely valid for large landslides, meaning large mass activation. How would your method be applied to medium to small landslide occurrences?

Thank you for the valuable feedback. The larger landslide generated lowerfrequency seismic signals. When the frequency is down to < 0.1Hz, single force inversion and landslide force history methods can be used for the volume estimate and trajectory retrieving (Ekström and Stark, 2013; Chao et al., 2017). Then, the highfrequency (>1Hz) seismic signals can further be used to obtain information about volume estimation, physical process recognition, and source location. However, for medium to small landslides, they only generated seismic signals of over 1 Hz, which means that the result would lose the sliding direction from inversion analysis. The gap is derived from the methodology limitation of high-frequency Green's function.

For the practical application, we rely on the geohazard location (GeoLoc; section 3.3) to detect the seismic source continuously. When the result shows a convergence pattern (Chang et al., 2023), it indicates that the location is reliable, and the process then proceeds to landslide classification. While most studies have used spectrogram features to identify landslide sources, some have employed seismic parameters to classify landslide types (Provost et al., 2018). In this research, we built upon the findings of Chang et al. (2021), who summarized the relationship between physical processes and spectrogram features, which were validated using several video recordings.

Once the landslide signals and physical processes are retrieved, the relationship between seismic magnitude and landslide volume can be used to estimate landslide size. Le Roy et al. (2019) summarized regression models from six different regions, though these regressions vary depending on geological conditions. In Taiwan, we developed two regression models to estimate landslide volume—one based on local magnitude and the other using amplitude at the source (Chang et al., 2021). Suppose a dense seismic network with real-time data transmission is deployed in a landslide-prone area. In that case, high-frequency seismic signals (>1 Hz) can provide accurate landslide locations, details of the physical process, and an estimate of the landslide volume within minutes to an hour after the event. This process generally requires expert analysis to interpret the physical processes, after which preliminary results can be shared with geologists for further survey and validation.

Our previous work has referred to some aspects of the flexible application of lowand high-frequency seismic signals (Chang et al., 2024). Additionally, we have expanded the content in a newly added subsection 5.4, Lines 457-462.

**References**

- Chang, J.M., Chao, W.A., Chen, H., Kuo, Y.T., and Yang, C.M.: Locating rock slope failures along highways and understanding their physical processes using seismic signals, Earth Surf. Dynam., 9, 505–517, https://doi.org/10.5194/esurf-9-505-2021, 2021.
- Chang, J.M., Chao, W.A., Kuo, Y.T., Yang, C.M., Chen, H., and Wang, Yu.: Field experiments: How well can seismic monitoring assess rock mass falling? Eng. Geol., 323, 107211, https://doi.org/10.1016/j.enggeo.2023.107211, 2023.
- Chang, J. M., Kuo, Y. T., Chao, W. A., Lin, C. M., Lan, H. W., Yang, C. M., and Chen,
  H.: Landslide Warning Area Delineation through Seismic Signals and Landslide
  Characteristics: Insights from the Silabaku Landslide in Southern Taiwan. Seismol.
  Res. Lett., 95(5), 2986-2996, 2024.
- Chao, W.A., Wu, Y.M., Zhao, L., Chen, H., Chen, Y.G., Chang, J.M., and Lin, C.M.: A first near real-time seismology-based landquake monitoring system, Sci. Rep., 7, 43510, https://doi.org/10.1038/srep43510, 2017.
- Ekström, G., and Stark, C. P.: Simple scaling of catastrophic landslide dynamics. Science, 339(6126), 1416-1419, 2013.
- Le Roy, G., Helmstetter, A., Amitrano, D., Guyoton, F., and Le Roux-Mallouf, R.: Seismic analysis of the detachment and impact phases of a rockfall and application for estimating rockfall volume and free-fall height. J. Geophys. Res. Earth. Surf., 124(11), 2602-2622, 2019.
- Provost, F., Malet, J.-P., Hibert, C., Helmstetter, A., Radiguet, M., Amitrano, D., Langet, N., Larose, E., Abancó, C., Hürlimann, M., Lebourg, T., Levy, C., Le Roy, G., Ulrich, P., Vidal, M., and Vial, B.: Towards a standard typology of endogenous landslide seismic sources, Earth Surf. Dynam., 6, 1059-1088, https://doi.org/10.5194/esurf-6-1059-2018, 2018.

**# Reviewer 2**

The study presented by Chang et al. is a valuable contribution to the analysis of landslide failure using existing seismic stations, expanding the application limits of previous research. However, several aspects require further review and justification to ensure the accuracy and rigor of the results.

1. Although the authors have selected existing numerical models (SF) and explained them clearly, a more detailed evaluation of their suitability for the study area and previous validation under similar conditions is essential. This would involve assessing the potential biases and limitations of the models and providing a detailed justification for their selection. Additionally, incorporating a more comprehensive validation of landslides would help avoid biases and ensure result accuracy. Considering terms like "probable landslide" could also enhance clarity and precision. I tried to review the video in S1, but was not available.

Thank you for your comments. We acknowledge that landslides can have longer durations and recognize the potential for underestimation of force magnitude when using a frequency band that does not encompass the entire duration of the landslide. However, it's pertinent to note that the lower frequency band is susceptible to influences from ocean waves in Taiwan. So, the selected frequency band in our study was inherited from the local experience of parameter setting in Chao et al. (2017), where the setting can fit most landslides in Taiwan for continuous monitoring and detection for more than 7 years. We further examined another parameter, source depth, which also does not influence the result. The relevant research was done in this year (Chang et al., 2024).

Further, the foundation for seismic source location using high-frequency signals was established by Chen et al. (2013), and this was later expanded by Chang et al. (2021), who integrated the "Amplitude source location" technique as the geohazard location system (GeoLoc). Chang et al. (2023) then refined the process by defining a set of rules to ensure reliable results. These efforts represent a significant investment in developing algorithms that are suitable for real-world application, especially in Taiwan's complex terrain.

This study focused on providing a timely and reliable assessment of landslide information for geologists. Because Taiwan is a narrow island. Every provincial highway is important for transportation. When a landslide occurred, the highway bureau would remove the landslide material as soon as possible. As the case when we conducted the field survey three days after the Cilan Landslide, the excavator had already worked to clean the road (Figure 8). The phenomenon brings obstacles to effective geological surveys after landslides. Therefore, how to give timely, moderate information to geologists is the key message in the study. Given the time-sensitive nature of landslide events, we utilized mutual techniques and parameters that have proven effective in Taiwan. This approach ensures that the system can deliver actionable data quickly. The relevant content has been added in subsection 5.4.

We reviewed all the web links in response to the issue. All are available. Please try again. Thank you!

**References**

- Chang, J.M., Chao, W.A., Chen, H., Kuo, Y.T., and Yang, C.M.: Locating rock slope failures along highways and understanding their physical processes using seismic signals, Earth Surf. Dynam., 9, 505–517, https://doi.org/10.5194/esurf-9-505-2021, 2021.
- Chang, J.M., Chao, W.A., Kuo, Y.T., Yang, C.M., Chen, H., and Wang, Yu.: Field experiments: How well can seismic monitoring assess rock mass falling? Eng. Geol., 323, 107211, https://doi.org/10.1016/j.enggeo.2023.107211, 2023.
- Chang, J. M., Kuo, Y. T., Chao, W. A., Lin, C. M., Lan, H. W., Yang, C. M., & Chen, H.: Landslide Warning Area Delineation through Seismic Signals and Landslide Characteristics: Insights from the Silabaku Landslide in Southern Taiwan. Seismol. Res. Lett., 95(5), 2986-2996, 2024.
- Chao, W.A., Wu, Y.M., Zhao, L., Chen, H., Chen, Y.G., Chang, J.M., and Lin, C.M.: A first near real-time seismology-based landquake monitoring system, Sci. Rep., 7, 43510, https://doi.org/10.1038/srep43510, 2017.

2. The detection of individual landslide events is a critical aspect of the study. To ensure the accuracy of the results, it is crucial to verify that detected events are individual and not the result of superimposed seismic signals. The authors could provide evidence to support that only one landslide is being observed at each time step, thereby strengthening the reliability of the findings.

Thank you for your valuable suggestion. To address the concern regarding detecting individual landslide events, we examine the spatial distribution of landslides to ensure no other significant landslides occurred in the vicinity during the study period. The results are shown in Figure A, which illustrates that landslides triggered by the typhoon were concentrated within the area depicted in Figure 1. If additional landslide sources had influenced the target signals, they would likely have originated from this specific region. However, most of the landslides shown in Figure 1 are located adjacent to the provincial highway, with their occurrence times roughly documented by the Directorate General of Highways. This data also confirms the Cilan Landslide is the more substantial one, whereas others are with a relatively smaller scale. Please see Line 104 and Figure S1 in the revised version.

Furthermore, when reviewing the high-frequency spectrograms in Figure 4, we

observed four continuous distinct V-shaped patterns, suggesting similar motion behavior. This clear and consistent pattern hints that the signals are most likely from a single landslide event. Please see Lines 251-252.

---

## Author Comment (AC2)

**Responses to Reviewer's Comments**

**Note**: Reviewer's comments are all quoted in their entirety and are in *Italics*, while authors' responses are in blue.

**Reviewer 1**
*I found your paper very interesting, especially from the fact that you can use seismology data to recreate the slope movement and direction. I cannot say I have anything to add to the paper.*
*The only one thing I would like to address is the use of the method for the landslide size and type. From the data presented in the paper, it looks like the method is completely valid for large landslides, meaning large mass activation. How would your method be applied to medium to small landslide occurrences?*

Thank you for the valuable feedback. The larger landslide generated lower-frequency seismic signals. When the frequency is down to < 0.1Hz, single force inversion and landslide force history methods can be used for the volume estimate and trajectory retrieving (Ekström and Stark, 2013; Chao et al., 2017). Then, the high-frequency (>1Hz) seismic signals can further be used to obtain information about volume estimation, physical process recognition, and source location. However, for medium to small landslides, they only generated seismic signals of over 1 Hz, which means that the result would lose the sliding direction from inversion analysis. The gap is derived from the methodology limitation of high-frequency Green's function.

For the practical application, we rely on the geohazard location (GeoLoc; section 3.3) to detect the seismic source continuously. When the result shows a convergence pattern (Chang et al., 2023), it indicates that the location is reliable, and the process then proceeds to landslide classification. While most studies have used spectrogram features to identify landslide sources, some have employed seismic parameters to classify landslide types (Provost et al., 2018). In this research, we built upon the findings of Chang et al. (2021), who summarized the relationship between physical processes and spectrogram features, which were validated using several video recordings.

Once the landslide signals and physical processes are retrieved, the relationship between seismic magnitude and landslide volume can be used to estimate landslide size. Le Roy et al. (2019) summarized regression models from six different regions, though these regressions vary depending on geological conditions. In Taiwan, we developed two regression models to estimate landslide volume—one based on local magnitude and the other using amplitude at the source (Chang et al., 2021).

Suppose a dense seismic network with real-time data transmission is deployed in a landslide-prone area. In that case, high-frequency seismic signals (>1 Hz) can provide accurate landslide locations, details of the physical process, and an estimate of the landslide volume within minutes to an hour after the event. This process generally requires expert analysis to interpret the physical processes, after which preliminary results can be shared with geologists for further survey and validation.

Our previous work has referred to some aspects of the flexible application of low- and high-frequency seismic signals (Chang et al., 2024). Additionally, we have expanded the content in a newly added subsection 5.4, Lines 457-462.

References

Chang, J.M., Chao, W.A., Chen, H., Kuo, Y.T., and Yang, C.M.: Locating rock slope failures along highways and understanding their physical processes using seismic signals, Earth Surf. Dynam., 9, 505–517, https://doi.org/10.5194/esurf-9-505-2021, 2021.

Chang, J.M., Chao, W.A., Kuo, Y.T., Yang, C.M., Chen, H., and Wang, Yu.: Field experiments: How well can seismic monitoring assess rock mass falling? Eng. Geol., 323, 107211, https://doi.org/10.1016/j.enggeo.2023.107211, 2023.

Chang, J. M., Kuo, Y. T., Chao, W. A., Lin, C. M., Lan, H. W., Yang, C. M., and Chen, H.: Landslide Warning Area Delineation through Seismic Signals and Landslide Characteristics: Insights from the Silabaku Landslide in Southern Taiwan. Seismol. Res. Lett., 95(5), 2986-2996, 2024.

Chao, W.A., Wu, Y.M., Zhao, L., Chen, H., Chen, Y.G., Chang, J.M., and Lin, C.M.: A first near real-time seismology-based landquake monitoring system, Sci. Rep., 7, 43510, https://doi.org/10.1038/srep43510, 2017.

Ekström, G., and Stark, C. P.: Simple scaling of catastrophic landslide dynamics. Science, 339(6126), 1416-1419, 2013.

Le Roy, G., Helmstetter, A., Amitrano, D., Guyoton, F., and Le Roux-Mallouf, R.: Seismic analysis of the detachment and impact phases of a rockfall and application for estimating rockfall volume and free-fall height. J. Geophys. Res. Earth. Surf., 124(11), 2602-2622, 2019.

Provost, F., Malet, J.-P., Hibert, C., Helmstetter, A., Radiguet, M., Amitrano, D., Langet, N., Larose, E., Abancó, C., Hürlimann, M., Lebourg, T., Levy, C., Le Roy, G., Ulrich, P., Vidal, M., and Vial, B.: Towards a standard typology of endogenous landslide seismic sources, Earth Surf. Dynam., 6, 1059-1088, https://doi.org/10.5194/esurf-6-1059-2018, 2018.

**Reviewer 2**

*The study presented by Chang et al. is a valuable contribution to the analysis of landslide failure using existing seismic stations, expanding the application limits of previous research. However, several aspects require further review and justification to ensure the accuracy and rigor of the results.*

*1. Although the authors have selected existing numerical models (SF) and explained them clearly, a more detailed evaluation of their suitability for the study area and previous validation under similar conditions is essential. This would involve assessing the potential biases and limitations of the models and providing a detailed justification for their selection. Additionally, incorporating a more comprehensive validation of landslides would help avoid biases and ensure result accuracy. Considering terms like "probable landslide" could also enhance clarity and precision. I tried to review the video in S1, but was not available.*

Thank you for your comments. We acknowledge that landslides can have longer durations and recognize the potential for underestimation of force magnitude when using a frequency band that does not encompass the entire duration of the landslide. However, it's pertinent to note that the lower frequency band is susceptible to influences from ocean waves in Taiwan. So, the selected frequency band in our study was inherited from the local experience of parameter setting in Chao et al. (2017), where the setting can fit most landslides in Taiwan for continuous monitoring and detection for more than 7 years. We further examined another parameter, source depth, which also does not influence the result. The relevant research was done in this year (Chang et al., 2024).

Further, the foundation for seismic source location using high-frequency signals was established by Chen et al. (2013), and this was later expanded by Chang et al. (2021), who integrated the "Amplitude source location" technique as the geohazard location system (GeoLoc). Chang et al. (2023) then refined the process by defining a set of rules to ensure reliable results. These efforts represent a significant investment in developing algorithms that are suitable for real-world application, especially in Taiwan's complex terrain.

This study focused on providing a timely and reliable assessment of landslide information for geologists. Because Taiwan is a narrow island. Every provincial highway is important for transportation. When a landslide occurred, the highway bureau would remove the landslide material as soon as possible. As the case when we conducted the field survey three days after the Cilan Landslide, the excavator had already worked to clean the road (Figure 8). The phenomenon brings obstacles to effective geological surveys after landslides. Therefore, how to give timely, moderate information to geologists is the key message in the study. Given the time-sensitive

nature of landslide events, we utilized mutual techniques and parameters that have proven effective in Taiwan. This approach ensures that the system can deliver actionable data quickly. The relevant content has been added in subsection 5.4.

We reviewed all the web links in response to the issue. All are available. Please try again. Thank you!

References

Chang, J.M., Chao, W.A., Chen, H., Kuo, Y.T., and Yang, C.M.: Locating rock slope failures along highways and understanding their physical processes using seismic signals, Earth Surf. Dynam., 9, 505–517, https://doi.org/10.5194/esurf-9-505-2021, 2021.

Chang, J.M., Chao, W.A., Kuo, Y.T., Yang, C.M., Chen, H., and Wang, Yu.: Field experiments: How well can seismic monitoring assess rock mass falling? Eng. Geol., 323, 107211, https://doi.org/10.1016/j.enggeo.2023.107211, 2023.

Chang, J. M., Kuo, Y. T., Chao, W. A., Lin, C. M., Lan, H. W., Yang, C. M., & Chen, H.: Landslide Warning Area Delineation through Seismic Signals and Landslide Characteristics: Insights from the Silabaku Landslide in Southern Taiwan. Seismol. Res. Lett., 95(5), 2986-2996, 2024.

Chao, W.A., Wu, Y.M., Zhao, L., Chen, H., Chen, Y.G., Chang, J.M., and Lin, C.M.: A first near real-time seismology-based landquake monitoring system, Sci. Rep., 7, 43510, https://doi.org/10.1038/srep43510, 2017.

*2.  The detection of individual landslide events is a critical aspect of the study. To ensure the accuracy of the results, it is crucial to verify that detected events are individual and not the result of superimposed seismic signals. The authors could provide evidence to support that only one landslide is being observed at each time step, thereby strengthening the reliability of the findings.*

Thank you for your valuable suggestion. To address the concern regarding detecting individual landslide events, we examine the spatial distribution of landslides to ensure no other significant landslides occurred in the vicinity during the study period. The results are shown in Figure A, which illustrates that landslides triggered by the typhoon were concentrated within the area depicted in Figure 1. If additional landslide sources had influenced the target signals, they would likely have originated from this specific region. However, most of the landslides shown in Figure 1 are located adjacent to the provincial highway, with their occurrence times roughly documented by the Directorate General of Highways. This data also confirms the Cilan Landslide is the more substantial one, whereas others are with a relatively smaller scale. Please see Line 104 and Figure S1 in the revised version.

Furthermore, when reviewing the high-frequency spectrograms in Figure 4, we

observed four continuous distinct V-shaped patterns, suggesting similar motion behavior. This clear and consistent pattern hints that the signals are most likely from a single landslide event. Please see Lines 251-252.

[Figure]

Figure A Map of the Landslide distribution triggered by Typhoon Nesat.

*3.    In Section 4.2, correlating landslide events and explicitly locating them on the area's geology and soils is vital. This would enable the assessment of the accuracy of the used density and avoid biases in mass/volume estimation. Furthermore, correlating events could facilitate the identification of patterns and trends in landslide occurrence, ultimately contributing to more effective prediction and prevention strategies. I would suggest increase the discussion using https://doi.org/10.29382/eqs-2020-0034*

Thank you for the comments. We understand the limitations of the seismic technique in estimating the landslide volume, like the analyzed frequency band, point source assumption, and data quality. Previous research has documented these limitations well (Allstadt et al., 2013; Li, et al., 2018). Therefore, we always use the terms "possible" or "roughly" to address relevant content (Lines 189 and 205). The primary role of seismic analysis is to provide pre-survey knowledge for timely assessment, while more precise mass or volume estimations rely on detailed fieldwork. Moreover, even with field surveys, accurately estimating landslide mass or volume

remains challenging. Therefore, while seismic data offers valuable insight, it does not drastically alter the overall picture in terms of density.

For the second issue, the reference paper used low-frequency seismic signals to describe the subevent. However, they suggested that high-frequency seismic signals offer a clear view for the identification of landslide patterns. Here, we used the spectrogram features to explain the pattern of continuous failure, where the spectrogram features were validated by Chang et al. (2021) through multiple landslide videos. Then, constrain the source location using GeoLoc and show the pattern of particle motion to support the landslide sequence.

We also acknowledge the reviewer's offer of a nice paper. After careful review, we involved part of the concept in the discussion part. Please see Lines 447-456.

References
Agliardi, F., Crosta, G., and Zanchi, A.: Structural constraints on deep-seated slope deformation kinematics, Eng. Geol., 59(1-2), 83-102, https://doi.org/10.1016/S0013-7952(00)00066-1, 2001.
Chang, J.M., Chao, W.A., Chen, H., Kuo, Y.T., and Yang, C.M.: Locating rock slope failures along highways and understanding their physical processes using seismic signals, Earth Surf. Dynam., 9, 505–517, https://doi.org/10.5194/esurf-9-505-2021, 2021.
Li, W., Zhang, Y., Xu, Y., Zheng, X., Wang, R., Su, J., Yi, G., and Huang, Q.: Complex dynamics of repeating and river-blocking landslides in Jiangda during 2018, Earthq. Sci., 34(1), https://doi.org/10.29382/eqs-2020-0034, 3-14, 2021.

*4.    Although the usage of digital elevation models (DEMs) is a strength of the study, the comparison between LiDAR and Drone-derived DEMs requires careful evaluation. Considering control points can clarify potential biases in reported heights or the need for vertical bias correction. This would ensure that the DEMs accurately represent the study area's topography. I would suggest consider https://doi.org/10.1111/tgis.12819 in your assessment.*

In response to your valuable suggestions, we have included statements regarding the vertical root mean square error (RMSE) of both the LiDAR DEM and the drone-derived digital surface model (DSM) in Section 4.4. Please see Lines 231-235: "The LiDAR DEM was obtained from the Ministry of the Interior, Taiwan. The drone-derived DSM was produced using ground control points via the e-GNSS service (Virtual Reference Station Real-Time Kinematic technology) provided by the Ministry of the Interior, Taiwan. The vertical root mean square errors for the LiDAR DEM and the drone-derived DSM were found to be 0.5 m and 0.2 m, respectively."

*5.   While the study provides valuable insights into landslide failure in the selected study area, it would be beneficial to discuss the potential applications of the methodology in different environmental contexts. For instance, how would the approach perform in regions with varying geological formations, climate conditions, or vegetation cover? Exploring these questions could enhance the study's relevance and generalizability, ultimately contributing to a more comprehensive understanding of landslide dynamics. The authors may consider incorporating case studies or comparative analyses to demonstrate the versatility of their approach. Please, consider discuss https://doi.org/10.5194/nhess-21-3015-2021, https://doi.org/10.1016/j.jhydrol.2021.126294, https://doi.org/10.1007/s10346-023-02179-4*

Thank you for the valuable suggestions. Creating a new subsection to enhance the visibility of these points is an excellent idea. We have thoroughly reviewed the suggested references and have ensured that the new content aligns closely with the paper's key message. The newly added subsection is divided into three parts. The first discusses the importance of timely information for the landslide community. The second is how to face medium- to small-scale landslides. Finally, we emphasize the importance of regular roadside slope inspections and how to respond effectively when damage signs are detected on slopes. Please see Section 5.4.